# Reversible Data Hiding Using Inter-Component Prediction in Multiview Video Plus Depth

**Jin Young Lee** [1,†] **, Cheonshik Kim** [2,*,†] **and Ching-Nung Yang** [3,†]

1    School of Intelligent Mechatronics Engineering, Sejong University, Seoul 05006, Korea;
     jinyounglee@sejong.ac.kr
2    Department of Computer Engineering, Sejong University, Seoul 05006, Korea
3    Department of Computer Science and Information Engineering, National Dong Hwa University,
     Hualien 97401, Taiwan; cnyang@gms.ndhu.edu.tw
*    Correspondence: mipsan@paran.com
†    These authors contributed equally to this work.

**Abstract:**    With the advent of 3D video compression and Internet technology, 3D videos have been deployed worldwide. Data hiding is a part of watermarking technologies and has many capabilities. In this paper, we use 3D video as a cover medium for secret communication using a reversible data hiding (RDH) technology. RDH is advantageous, because the cover image can be completely recovered after extraction of the hidden data. Recently, Chung et al. introduced RDH for depth map using prediction-error expansion (PEE) and rhombus prediction for marking of 3D videos. The performance of Chung et al.'s method is efficient, but they did not find the way for developing pixel resources to maximize data capacity. In this paper, we will improve the performance of embedding capacity using PEE, inter-component prediction, and allowable pixel ranges. Inter-component prediction utilizes a strong correlation between the texture image and the depth map in MVD. Moreover, our proposed scheme provides an ability to control the quality of depth map by a simple formula. Experimental results demonstrate that the proposed method is more efficient than the existing RDH methods in terms of capacity.

**Keywords:** 3D; depth map; inter-component prediction; MVD; reversible data hiding; texture

## 1. Introduction

Data hiding (DH) [1] plays an important role in secret communication. For this purpose, secret information and metadata are embedded in the cover media, such as still image, video, audio, 3D video, and so on. The visual quality and capacity of the cover image are important criteria for DH schemes. In addition, reversible DH (RDH) techniques are developed to extract the embedded secret information and restore losslessly the cover image.

Up to date, various RDH algorithms have been proposed, e.g., difference expansion-based algorithms [2–6], histogram shifting [7–10], prediction-error expansion (PEE) [11–14] and integer-to-integer transform [15–17], etc.

The approaches of difference expansion (DE) show good performance in respect of high-capacity. The first introduction of this algorithm was by Tian, and the research has been extended by [3,4]. For data embedding, it has to make a room for a secret bit through a pixel extension, and inserts data therein. Alattar [3] improved the performance of Tian's work by generalizing a DE technique for all integer conversions. The method proposed by Sachnev et al. is that image pixels are separated into black and white squares of a chessboard with two identical sets diagonally connected. This prediction method, called rhombus, is superior to the existing prediction methods (e.g., median edge detector

predictor (MED) and gradient-adjusted predictor (GAP)) by making the average value of adjacent neighbors of a specific pixel as a predictive value. Thereafter, various methods for improving prediction performance were introduced.

The histogram shifting (HS) technique is also known as a method having relatively a few distortion in a cover image. However, it requires a location map in RDHs to embed data and restore a cover image. Al-Qershi and Khoo proposed a 2-dimensional DE (2D-DE) scheme achieving about 1-BPP performance [6]. These histogram-based schemes may achieve good visual quality and adequate embedding capacity, but it has the drawback having to send a pair of peaks and zero points to the receiver.

PEE involves a process that obtains the prediction error (PE) from the neighborhood of a pixel and embeds information bits into the expanded errors. If the difference between an original pixel and a predicted pixel is large, the distortion of the cover image is greatly enlarged during the embedding process. In this case, by enhancing the data embedding capacity of the region of low frequency in the cover image, it may maintain the image quality and embedding capacity of the cover image. Compared to DE and HS-based methods, it is well known that PEE performs better. When we are considering the existing PEE methods, DH with order prediction has less distortion at low embedding rates.

Meanwhile, with the rapid development of multiview video technologies, viewers can experience more realistic 3D scenes with highly advanced multimedia systems, such as 3D television and free-viewpoint television. To overcome a limited bandwidth, the multiview video plus depth (MVD) is adopted as a 3D video format [18,19]. In MVD (see Figure 1), a texture image indicates intensities of an object, whereas a depth map represents a distance between an object and a camera as a grayscale image having values between 0 and 255. Because MVD enables the advanced video system to arbitrarily generate virtual views by using a depth image-based rendering (DIBR) method [20], a small number of view information can be transmitted.

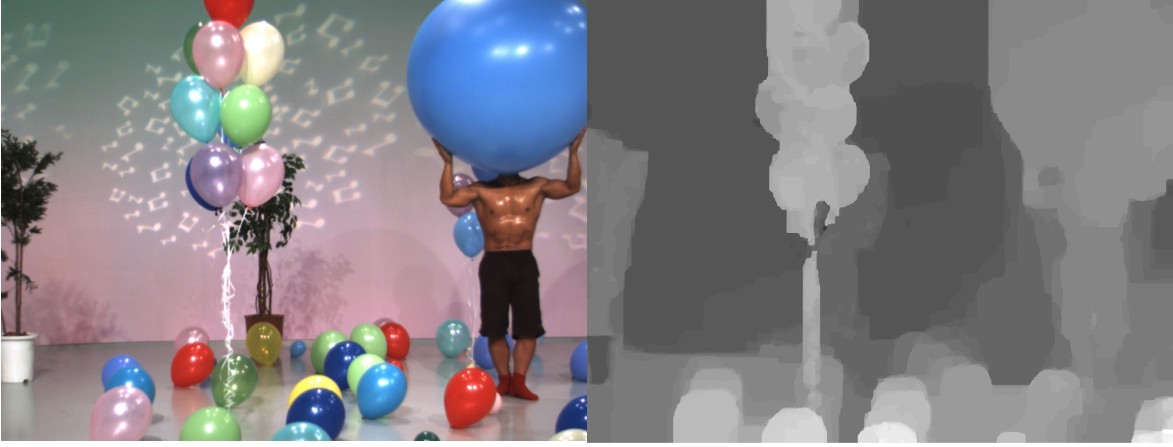

**Figure 1.** MVD consisting of a texture image (**left**) and its corresponding depth map (**right**).

Until now, various watermarking technologies [21–25] have been introduced for marking 3D videos. Asikuzzaman and Pickering [21] proposed a digital watermarking approach that inserts a watermark into the DT-CWT coefficients. Pei and Wang [22] introduced a 3D watermarking technique based on the D-NOSE model which can detect the suitable region of the depth image for watermark embedding. Since view synthesis is very sensitive to variations in-depth values, this scheme focuses mainly on the synthesis error. Wang et al. [23] exploited scale-invariant feature transform (SIFT)-based feature points to synchronize a watermark but focused on only signal processing and omitted geometric attacks.

Based on MVD, Chung et al. [26] and Shi et al. [27] proposed RDHs for depth maps using a depth no-synthesis-error (D-NOSE) model [28] and PEE. Each pixel in the depth maps has an allowable pixel range and the pixel value is increased or decreased in an allowable range. However, it may be

guaranteed that there are no errors in the synthesized image using D-NOSE. Taking advantage of this characteristic may be used to improve embedding capacity for RDH. Chung et al. first proposed a method based on PEE that could effectively hide data in depth maps of 3D images. The disadvantage of Chung et al.'s method is that it does not provide sufficient embedding capacity. Shi et al. proposed a way to use all of the acceptable range of pixels to solve the problem proposed by Chung et al.'s method. However, Shi et al. did not suggest a systematic way of adjusting the embedding capacity considering a quality of depth map.

In this paper, we first analyze the disadvantages of Chung et al.'s reversible data hiding algorithm and propose a PEE-based DH technique that completely uses the allowable range of each pixel using an inter-component prediction method. The performances of the proposed RDH are improved by using the correlation between the texture and the depth map, which is an advantage of the inter-component prediction. Also, we propose a method to control embedding rates and image quality systematically which may be applied to various RDH applications such as medical or military fields.

The remainder of this paper is organized as follows. Section 2 briefly discusses view synthesis, difference expansion, related RDH and watermarking methods. In Section 3, we introduce a reversible data hiding method based on D-NOSE model, PEE, and inter-component prediction. In Section 4, we compare and analyze the experimental result with conventional RDHs and our proposed RDH. Finally, this paper concludes in Section 5.

## 2. Related Works

In Section 2, we explain a 3D view synthesis principle, a difference expansion (DE) method, and Chung et al.'s [26] and Shi et al.'s [27] RDH methods based on 3D view synthesis. In addition, Zhu et al.'s [24], Wang et al.'s [23], and Asikuzzaman et al.'s [25] digital watermarking methods are also introduced.

### 2.1. View Synthesis

We can obtain a 3D version of the classical 2D videos with depth information via 3D view synthesis. Depth information plays a key role in synthesizing virtual views and the quality of synthesized views is critical in 3D video systems. In view synthesis, a pixel in the texture image is mapped to a new position in the virtual view by using the corresponding depth value. First, the disparity $d$ in a pixel of depth map is obtained using the following equation.

$$d_i = \frac{f \cdot l}{255} \left( \frac{1}{z_{near}} - \frac{1}{z_{far}} \right) \times q_i. \tag{1}$$

where $f$ and $l$ denote the focal length and the baseline distance between two horizontally adjacent cameras, respectively, and $z_{near}$ and $z_{far}$ mean the nearest and farthest depth values, respectively. The pixel $q$ indicates the $i_{th}$ depth pixel value. After the calculation of a disparity $d_i$, it is rounded into an integer value. Then, the pixel $(x', y')$ is filled by shifting the pixel $(x, y)$ with $d$ (see Figure 2).

$$\begin{pmatrix} x' \\ y' \end{pmatrix} = \begin{pmatrix} x \pm d \\ y \end{pmatrix} \tag{2}$$

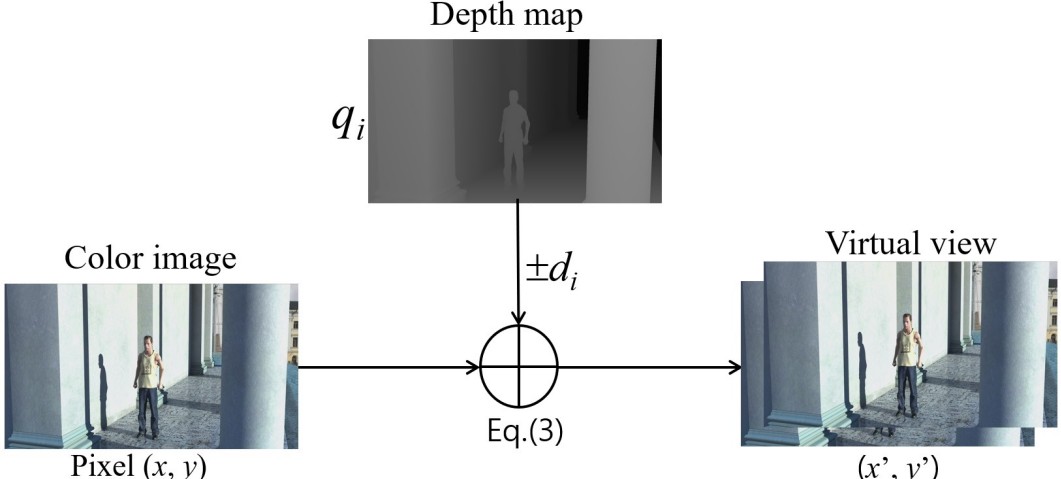

**Figure 2.** View synthsis based on disparity information.

Based on the D-NOSE model [28], the symbols $L$ and $U$ indicate the minimum and maximum depth values within the allowable distortion range. If the marked pixel $q'_i$ is still in the range ($[L, U]$) after hiding the data in $q_i$, the virtual view will not be distorted. The notation $\varphi$ is a set collecting depth pixel value with the disparity ($d_i = n$). $N$ denotes the number of pixels (width $\times$ height) in the depth map.

For example, assuming that a disparity $n = 32$ and the minimum and maximum pixels belonging to $n$ are $[L_q, U_q] = \{21...24\}$. It means that there are four pixels corresponding to disparity $d_i = 32$, i.e., the pixels are $21, \ldots, 24$. The way to figure out the allowable range for each pixel of the depth map is shown stepwise in the Algorithm 1.

$$\varphi_n = \{q_i \in (d_i = n)\}_1^N \tag{3}$$

---

**Algorithm 1** DRange($q$)

---

Input: input pixel $q$
Output: $[L, U]$

1: input pixel $q$
2: $d \leftarrow$ depth2disprity(q) // Equation (1)
3: **for** $i = q - 1$ To 0 Step $-1$ **do**
4:     **if** depth2disprity(i)! $= d$ **then** break
5:     **end if**
6: **end for**
7: $L \leftarrow i + 1$
8: **for** $i = q + 1$ To 255 Step 1 **do** //8-bit depth pixel
9:     **if** depth2disprity(i)! $= d$ **then** break
10:     **end if**
11: **end for**
12: $U \leftarrow i - 1$
13: **return** [L, U]

---

### 2.2. Difference Expansion

In this section, we describe the concept of RDH using pixel prediction and DE, where $p$ and $\hat{p}$ denote an original pixel and a predicted pixel. A prediction error is determined by $e_{i,j} = p_{i,j} - \hat{p}_{i,j}$. If $e_{i,j} < T$ and no overflow and underflow on each pixel, the secret bits $b$ may be embedded into the pixel

$p$ as $p'_{i,j} = p_{i,j} + e_{i,j} + b$. If $|e_{i,j}| \geq T$, it is not appropriate to embed secret bits, because the carrier pixel $p'$ may have higher prediction error than the other embedded pixels. This pixel is modified as follows:

$$p'_{i,j} = \begin{cases} p_{i,j} + T, & \text{if } (e_{i,j} \geq T) \\ p_{i,j} - (T-1), & \text{if } (e_{i,j} \leq -T) \end{cases}$$

The location information of the underflow or overflow is recorded in the location map and is used for decoding. If the predictive values are the same before and after data hiding, the reversibility of the watermarking is guaranteed. It should restore the shifted values $T$ before obtaining the predictive value. Then, it is possible to obtain the predictive values correctly.

### 2.3. Chung et al.'s Reversible Data Hiding

The D-NOSE model can guarantee zero synthesis distortion by determining an allowable distortion range for each depth pixel. The previous two existing methods [26,27] use the rhombus prediction to obtain the prediction pixel $\hat{p} = \left\lceil \frac{p_1 + p_2 + p_3 + p_4}{4} \right\rceil$, and the prediction error $e = p - \hat{p}$ means the difference between the original pixel $p$ and the predicted depth pixel $\hat{p}$. Here, $p_1, p_2, p_3$, and $p_4$ denote the four adjacent pixels, where they are placed on the top, left, bottom, and right sides of $p$, respectively. The average value is rounded up in the calculation. If the number of hidden bits $m$ for a pixel $p$ is $\lfloor log_2(U - \hat{p}_i + 1) \rfloor$ when $e = 0$ and $m = \lfloor log_2(p_i - L + 1) \rfloor - 1 \rfloor$ when $e = -1$. Otherwise, $m = 0$.

The marked pixel $p'$ is obtained from $p' = \hat{p} + b$ when $e = 0$ and $p' = \hat{p} - b - 1$ when $e = -1$. Here, $b$ means a binary number of $m$ bits, and the allowable range is $[L, U]$. On the receiving side, the secret bits $b$ can be simply obtained by the expression $b = (e' \bmod 2^m)$, where $e' = p' - \hat{p}$.

The location map in Chung et al.'s method is a simple way, i.e., mark "1" if there is no hidden data at the position and "0" otherwise. If the surface of the image is the same color, most of the location map may be zeros since most of prediction errors may be $e = 0$. For this reason, Chung et al. used arithmetic coding to compress the location map and then hide the location map in the front or back end of the depth map. Chung et al.'s method achieves the purpose of RDH for 3D synthesis images. Unfortunately, it does not embed a sufficient amount of data.

There is a vulnerability in Chung et al.'s method. For example, if a pixel $p$ and an error $e$ is 85 and 0, respectively, and acceptable range of the pixel $p$ is [83, 87], then it is allowed to hide only 1-bit in the pixel $p$, because of $m = \lfloor log_2(87 - 85 + 1) \rfloor = 1$. If a pixel is $p = 87$ and $e = 0$, then it does not allow to embed bits. That is because a room to hide bits is determined by the position of the predicted pixel in an allowable range of a pixel. Thus, there is no room to hide a bit when the pixel is $p = 87$. Another vulnerability is that the quality of the depth map is not considered sufficiently, because the quality only depends on the feature of the depth map.

### 2.4. Shi et al.'s Reversible Data Hiding

Shi et al. [27] proposed a RDH based on D-NOSE model, where the method embed information into double layer of depth maps. Here, the prediction method for PEE obtains $\hat{p}$ by the method of rhombus prediction. In order to use of the allowable range fully, it is embed the data into the prediction-error ($e = p - \hat{p}$) value 0, and the pixel is expanded toward either the maximum or the minimum values. In the allowable range $[l_n, u_n]$ of the disparity $d_i = n$ in the pixel, the number of bits for the pixel $q_i$ is $m_i = log_2(u_n^* - l_n^* + 1)$ where $\hat{q}_i \leq u_n^* \leq u_n$ and $l_n \leq l_n^* \leq \hat{q}_i$ when $e_i = 0$. Otherwise, it is $m_i = 0$. The marked pixel $q'_i$ may be expressed as $q'_i = l_n^* + b$, where $b = \{0, 1, 2, \ldots, 2^m - 1\}$. For example, if a pixel $p$ and an error $e$ is 85 and 0, respectively, and acceptable range of the pixel $p$ is [83, 87], then it is allowed to hide $m = log_2 5$.

### 2.5. Zhu et al.'s Digital Watermarking

Zhu et al. [24] propose a watermarking method for a new viewpoint video frame generated by DIBR (Depth Image-Based Rendering). To preserve the watermark information during the generation of the viewpoint video frame, the blocks in foreground object of original video frame is selected to embed the watermark because pixels in this kind of object are more likely to be preserved in the warping. Here, for watermarking, DCT transformation of the blocks in foreground object is firstly done. Then, after embedding the watermark into the DCT, IDCT should be done before the DIBR.

### 2.6. Wang et al.'s Digital Watermarking

Wang et al. [23] propose a novel watermarking method for DIBR 3D images by using SIFT to select the area where watermarking should be embedded. Then, the watermark information is embedded into the DCT coefficients of the selected area by using Spread spectrum technology. The SIFT is used to select suitable areas in which watermarking should be embedded by applying $n \times n$ 2D-DCT to the areas selected. Next, spread spectrum technique and orthogonal spread spectrum code are applied to embed the watermark. In order to extract watermarks from images, we can compute the correlation between DCT coefficients of every area and the spread spectrum code to estimate the embedded message.

### 2.7. Asikuzzaman et al.'s Digital Watermarking

Asikuzzaman et al. [25] proposed a blind video watermarking algorithm in which a watermark is embedded into two chrominance channels using a double tree complex wavelet transform. The chrominance channel has a watermark and preserves the original video quality and the double tree composite wavelet transform ensures robustness against geometric attacks due to the shift invariant nature. The watermark is extracted from a single frame without the original frame. This approach is also robust to downscaling in arbitrary resolution, aspect ratio change, compression, and camcording.

## 3. Proposed Scheme

In this section, we introduce RDH based on the D-NOSE model by using PEE and inter-component prediction to improve the accuracy of predictive pixel in our proposed method. The accuracy of the prediction maximizes the performance of the proposed RDH based on the D-NOSE model using maximum allowable pixel range of each pixel. Moreover, our proposed scheme has the capability controlling the quality and embedding capacity of the depth map.

RDH methods based on the 2D texture images usually have a trade-off between the capacity and quality of the cover signal. However, since the depth map is used to synthesize virtual views as non-visual data, we may embed secret data into the depth map without degrading the quality of the virtual view synthesis. The D-NOSE model can guarantee zero synthesis distortion by determining an allowable distortion range for each depth pixel. Besides, since we apply the existing PEE technology to the 3D depth map and use the location map, our proposed method is also a perfect way to restore the original depth map.

### 3.1. Inter-Component Prediction

Here, we will introduce inter-component prediction to obtain a better prediction for high embedding capacity. Figure 3 illustrates the configuration of the depth pixel, and inter-component prediction based on the corresponding texture pixels. A depth map in Figure 3a is composed of marked '♦' and '◇'. The pixels in depth map are subdivided into two sets, $\Phi 1 (\in \{ \blacklozenge \ldots \blacklozenge \})$ and $\Phi 2 (\in \{ \lozenge \ldots \lozenge \})$. The inter-component prediction (in Figure 3b) selects adaptively the predicted direction for the depth pixel by comparing the corresponding texture pixel and its adjacent pixels.

When the depth pixel $q_{i,j}$ is predicted from neighbor pixels, $q_1$, $q_2$, $q_3$, and $q_4$, as shown in Figure 3b, the corresponding texture pixel $t_{i,j}$ is compared with $t_1, t_2, t_3$, and $t_4$ as follows

$$dirD = \min_{dirT}(|t_1 - t|, |t_2 - t|, |t_3 - t|, |t_4 - 4|), \tag{4}$$

where $dirT$ and $dirD$ denote the prediction direction of the texture image and the depth maps, respectively. For example, if the pixel having the minimum texture difference is $t_1$, $q_{dirD}$ is set as $q_1$. The existing rhombus prediction method was an efficient prediction method for color and grayscale images. However, the rhombus method is less effective than the proposed method for depth map pixels. That is why we propose a way to predict the pixel of depth map by considering the texture image. Obviously, considering 3D synthesis, it seems that inter-component prediction is an excellent method.

The prediction is alternately performed for two sets, Φ1 and Φ2. That is, Φ2 is used to predict Φ1, and vice versa. When predicting Φ2, the prediction may not be accurate because there are hidden pixels in Φ1. To tackle this matter, we use the average of the unmarked pixel in Φ1. This is because the correlation between the marked depth pixels and the associated pixels is low, and the prediction is not accurate.

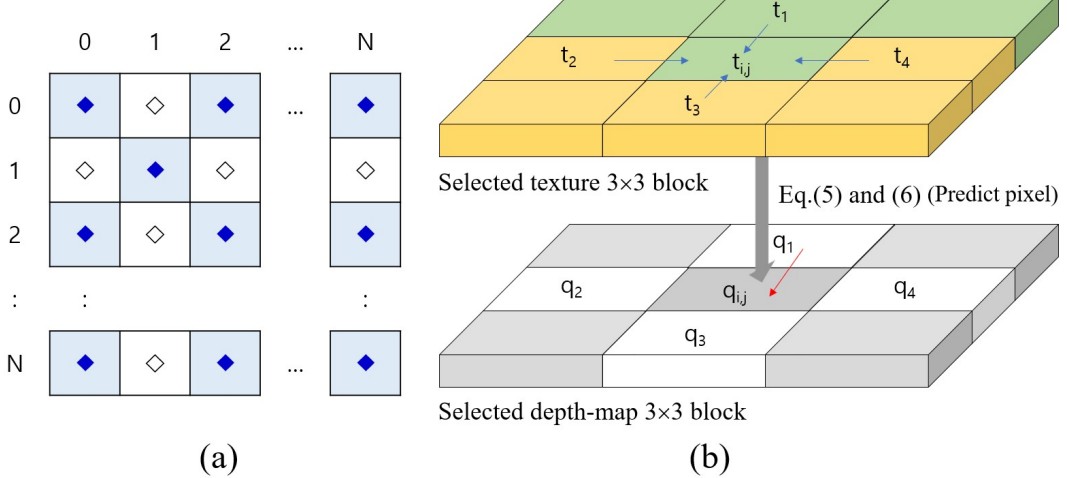

(a)　　　　　　　　　　　　　　　　　(b)

**Figure 3.** Diagram for configuration of the depth map, and inter-component prediction based on the corresponding texture pixels.

### 3.2. Embedding Algorithm

Here, we will examine in detail the data embedding procedure. To extract hidden bits and recover the original depth map, we have to record locations on whether each pixel contains hidden bits or not.

**Step 1:** Reads $3 \times 3$ block from the given depth map and assigns it to the variable $B$. Obtain the pixel $q_{i,j}$ and predicted pixel $q_{dirD}$ from $B$ using Equation (4).

**Step 2:** The prediction error $e$ is calculated by subtracting the prediction pixel $q_{dirD}$ from the original pixel $q_{i,j}$ as follow

$$e_{i,j} = q_{i,j} - q_{dirD}. \tag{5}$$

**Step 3:** If ($e_{i,j} = 0$), the number of bits $m$ that can be embedded into the pixel $q_{i,j}$ is calculated using the allowable pixel range of the pixel $q_{i,j}$ (Equation (6)), where $(L_q, U_q)$ is obtained from the Algorithm 1.

$$\begin{cases} m = \lfloor log_2(U_q - L_q + 1) \rfloor. \\ \text{if } (m < 1), \text{ goto Step 1} \end{cases} \tag{6}$$

Here, $(\lfloor x \rfloor)$ is the integer less than or equal to $x$.

**Step 4:** The binary secret bits $\eta_m$ are embedded in $q_{i,j}$ using Equation (7). (Note: the function $b2d(\cdot)$ is to converts binary values to a decimal value). That is, the $\eta_m$ is included in the expanded $e'$ by DE.

$$q'_{i,j} = \begin{cases} e'_{i,j} = 2^m \times e_{i,j} + b2d(\eta_m) \\ L_q + e'_{i,j} \end{cases} \tag{7}$$

**Step 5:** if $(e_{i,j} = 0)$, $LM_{i,j} = 0$, otherwise $LM_{i,j} = 1$. (Notes: location map $LM_{i,j} = 0$ means that hidden bits exists.)

$$LM_{i,j} = \begin{cases} 0, & \text{if } (e_{i,j} = 0) \\ 1, & \text{otherwise} \end{cases} \tag{8}$$

**Step 6:** Go to Step 1 until all pixels are processed.

After the embedding procedure is finished, all pixels including the hidden bits are still within the acceptable range. Therefore, 3D synthesis images have no distortion.

**Example 1.** *Given two blocks in Figure 4a, we demonstrate how to hide secret bits in the pixel $q_{i,j}$. First, we obtain the predictive pixel from the texture block and depth block. Applying Equation (4), it is observed that $t_1 = 51$ is the optimum pixel, so $q_{dirD}$ becomes $q_{i-1,j} = 22$.*

After applying Equation (5), if $e_{i,j} = (q_{i,j} - q_{dirD}) = 22 - 22 = 0$, we may obtain disparity and allowable range through Algorithm 1. i.e., $[L_q, U_q] = (20, 23)$. The number of embedded bits in the pixel can be obtained via the Equation (6), i.e., $m = 2$. Thus, it takes 2-bits from the secret data and converts it into a decimal value. By applying the Equation (7), $q'_{i,j} = 20 + 2 = 22$. Finally, the pixel $q'$ has secret bits $'10'_2$. In this case, $q_{i,j}$ is the same as $q'_{i,j}$, so there is no noise in the marked pixel $q'_{i,j}$.

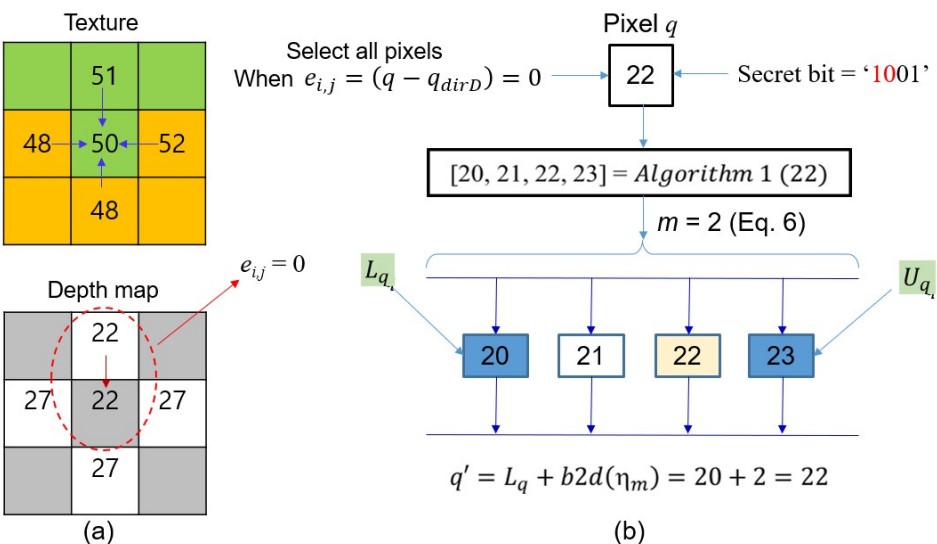

**Figure 4.** Example of the data embedding using the proposed method.

*3.3. Extraction Algorithm*

Suppose that a depth map containing secret data and a location map was delivered to the receiver side. We describe the procedure of extracting the hidden data and recover original pixels from the depth map. The detail (stepwise) is as follows.

**Step 1:** Reads $3 \times 3$ block from the given depth map and assigns it to the variable $B$. Obtains the pixel $q'_{i,j}$ and the predicted pixel $q_{dirD}$ from the $B$, respectively.

**Step 2:** If the location map $LM_{i,j} = 0$, we obtain the size of hidden bits $m = \lfloor (U_q - L_q + 1) \rfloor$ from the allowable range of $q_{i,j}$.

**Step 3:** Obtain the error $e'_{i,j}$ from the pixel $q'_{i,j}$ including secret bits by using Equation (9). (Notes: $d2b(x,m)$ converts the decimal number $x$ to a binary number of $m$ bits.)

$$\eta_m = \begin{cases} e'_{i,j} = q'_{i,j} - L_q \\ b = d2b(e'_{i,j}, m) \end{cases} \tag{9}$$

**Step 4:** The original pixel $q_{i,j}$ is restored by replacing the pixel $q'$ with $q_{dirD}$. Go to Step 1 until all pixels are processed.

**Example 2.** *First, Assuming that through the data embedding procedure (see Figure 4) in Example 1, the pixel $q'_{i,j}$ having the binary bits $'10'_2$ was transferred to the receiver. On the receiving side, some of the procedure for extracting the hidden bits in $q'_{i,j}$ are similar to the embedding procedure. The pixel $q_{dirD}$ is determined using the neighboring pixels of the pixel $q'_{i,j}$ and the inter-component prediction method. In this case, the predicted pixel is $q_{dirD} = 22$. Since the location map $LM_{i,j} = 0$ at position $(i,j)$ of $q'_{i,j}$, it can be seen that the data is hidden. Thus, the number of hidden bits from the allowable pixel range of the pixel $q'_{i,j}$ may be determined. That is, if we apply Equation (6) to the pixel $q'_{i,j}$, then we obtain the $m = \lfloor log_2(U_q - L_q + 1) \rfloor = \lfloor log_2(23 - 20 + 1) \rfloor = 2$. Next, the error bit $e'_{i,j} = 2$ obtains from $q'_{i,j} = 22$ and $L_q = 20$ by using Equation (9). The value hidden in the pixel $q_{i,j}$ is $e'_{i,j} = q'_{i,j} - L_q = 22 - 20 = 2$, which is converted to the binary number $'10'_2$. The marked pixel $q'_{i,j}$ is reconstructed to the original pixel $q_{i,j}$ by assigning the prediction value $q_{dirD}$ to the position $(i,j)$.*

### 3.4. Embedding/Extraction Procedure for Location Map

The location map (LM) is necessarily required for data extraction and depth map restoration, and thus, the location map should be transferred to the receiver side. There are many ways to transfer location map, but the most common way is to hide it into a cover image. At this step, it should minimize the location map, because the capacity of the secret data is reduced with the size of the location map. Thus, the compression of location map is a common procedure before embedding it into the cover image. There are several compression methods, but here we use arithmetic coding. The map size may be reduced by less than 10% by arithmetic coding, because the ratio of "0" in the map is more than 90%.

The location map $LM$ compressed by arithmetic coding is assigned into variable $\delta$. The compressed location data $\delta$ is embedded by the LSB replacement as follows, but the data are embedded considering the allowable pixel range.

$$q'_{i,j} = \begin{cases} \text{if } (U_q - L_q \geq 3) \ \{ \\ \quad q_{i,j} + 1, \quad \left. \begin{cases} \text{if } (\delta_{i,j} = 0 \ \& \ q_{i,j} \% 2 = 1) \\ \text{if } (\delta_{i,j} = 1 \ \& \ q_{i,j} \% 2 = 0) \end{cases} \right\} \equiv A \\ \quad q_{i,j} - 1, \quad \text{if } (q_{dirD} = U_q \ \& \ A) \\ \} \end{cases} \tag{10}$$

The compressed location data $\delta$ is embedded in front of the depth map. The data $\delta$ does not cause a synthesis error since it also adopted a D-NOSE model that hides the data within the allowable pixel range. The last position of the location information is sent on a separate channel. On the receiving side, the compressed location map can be extracted by the following Equation (11).

$$\delta_{i,j} = \begin{cases} \text{if } (U_q - L_q \geq 3) \ \{ \\ \quad 0, \quad \text{if } (q_{i,j} \% 2 = 0) \\ \quad 1, \quad \text{otherwise} \\ \} \end{cases} \tag{11}$$

### 3.5. Quanlity Control

In our proposed method, the quality of the depth map can be somewhat reduced if the allowable pixel range on all pixels is fully used. It can certainly be an advantage in terms of embedding capacity, but it is not desirable in terms of quality. Therefore, it is also important to find a balance between the two criteria. Adjustment of the allowable pixel range may be used to achieve such a purpose. Equations (12) and (13) can be used for managing depth map quality and embedding capacity through Equation (6). The control of depth map is achieved by using a limited allowable range $[L_q', U_q']$ instead of the $[L_q, U_q]$.

$$L_q' = \begin{cases} q_{dirD} - \sigma, & \text{if } (q_{dirD} - \sigma < L_q) \\ L_q, & \text{otherwise} \end{cases} \tag{12}$$

$$U_q' = \begin{cases} L_q' + 2^\sigma - 1, & \text{if } (L_q' + 2^\sigma - 1 > U_q) \\ U_q, & \text{otherwise} \end{cases} \tag{13}$$

Here, $\sigma$ is an integer variable and the range of values is $\{\sigma \geq 1 \,\&\, \sigma \leq n\}$. That is, increasing the value of $\sigma$ can increase the embedding capacity of RDH, while decreasing the value of $\sigma$ may improve the quality of the depth map. We may increase the usefulness of the proposed method if we adjust the value of $\sigma$ appropriately for the application.

## 4. Experimental Results

To better demonstrate the performance of our proposed scheme, we graphically show the results of experiments and analysis of 3D images with various features. All experiments are performed with eight 3D sized $1920 \times 1088$ (*or* $1024 \times 768$), "Poznan_Hall2","Poznan_Street", "Undo_Dancer", "GT_Fly", "Kendo", "Balloons", "Newspaper", and "Shark" (see Figure 5), which are often used for 3D video coding standards, such as 3D-AVC [19] and 3D HEVC [18], and the view synthesis reference software (VSRS) in JCT-3V [29]. Since the schemes such as Chung et al.'s, Shi et al.'s, and the proposed methods adopt the D-NOSE model, the synthesis using the original depth map is identical to that of the synthesis with the marked depth map.

In this paper, we use two criteria to evaluate the performance of the existing and our proposed schemes. The first criterion is the embedding rates (ERs) and the second is the peak signal-to-noise ratio (PSNR). The most well-known measurement method for objective evaluation of images is PSNR. That is, PSNR is the intensity of noise over the maximum intensity the signal can have. The MSE used in PSNR is an average difference in intensity between the marked depth map and the reference depth map. If the MSE value of the depth map is low, it is evaluated that the quality of the image is good. The MSE is calculated using a reference depth map $p$ and the distorted depth map $p'$ as follows.

$$MSE(p, p') = \frac{1}{N} \sum_{i=1}^{N} (p_i - p_i')^2 \tag{14}$$

The error value $\varepsilon = p_i - p_i'$ denotes the difference between the original depth map and the distorted depth map signal. The $255^2$ means the allowable pixel intensity in Equation (15).

$$PSNR = 10 log_{10} \frac{255^2}{MSE} \tag{15}$$

Meanwhile, ER is a measurement of how much information is included in the marked depth map. That is, ER is the ratio of the embedded information contained in the marked depth map. In Equation (16), $N$ is the total number of pixels and $||\eta||$ denotes the number of message bits.

$$\rho = \frac{||\eta||}{N} \tag{16}$$

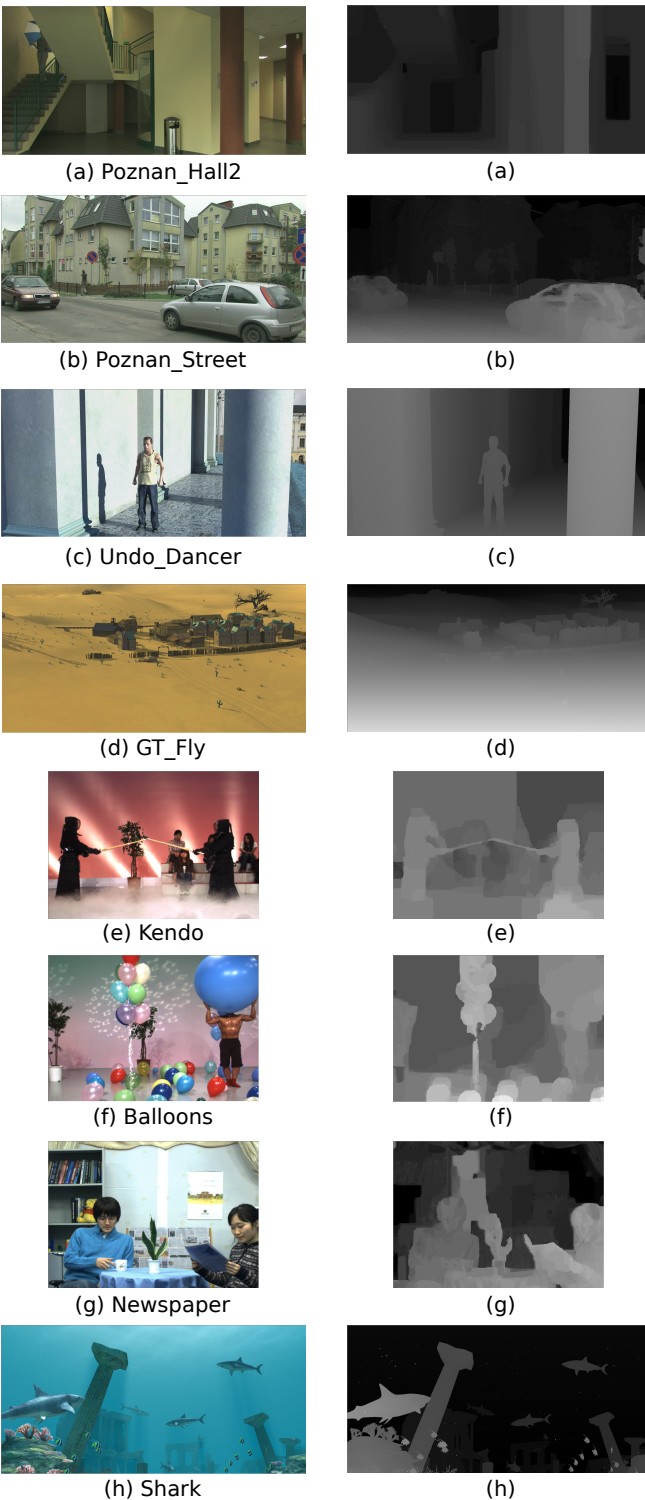

**Figure 5.** Test images; (**a**–**d**,**h**): 1920 × 1088, (**e**–**g**): 1024 × 768.

Figure 6 shows the comparison of data embedding rates using three methods (two existing methods and the proposed method) and eight depth maps: (a) Poznan_Hall2, (b) Poznan_Street, (c) Undo_Dancer, (d) GT_Fly, (e) Kendo, (f) Balloons, (g) Newspaper and (h) Shark; 1920 ×1088: (a)–(d) and (h), 1024 ×768: (e)–(g).

The ERs of Chung et al.'s method is much less than that of Shi et al.'s and our proposed methods. That is because Chung et al.'s method does not fully use the allowable range. On the other hand, Shi et al.'s and our proposed method may achieve better results by adopting an allowable pixel

range. For example, we show that the ERs of "Poznan_Hall2" in Chung et al.'s method is very low. For the "GT_Fly", the embedding rates of our proposed method is 0.22 BPP higher than that of Shi et al.'s method. As shown in Figure 6, the coefficients of difference between these two methods show that the performance of our inter-component prediction is superior to the rhombus prediction. Therefore, our proposed method outperforms the existing two methods, including Chang et al.'s and Shi et al.'s method.

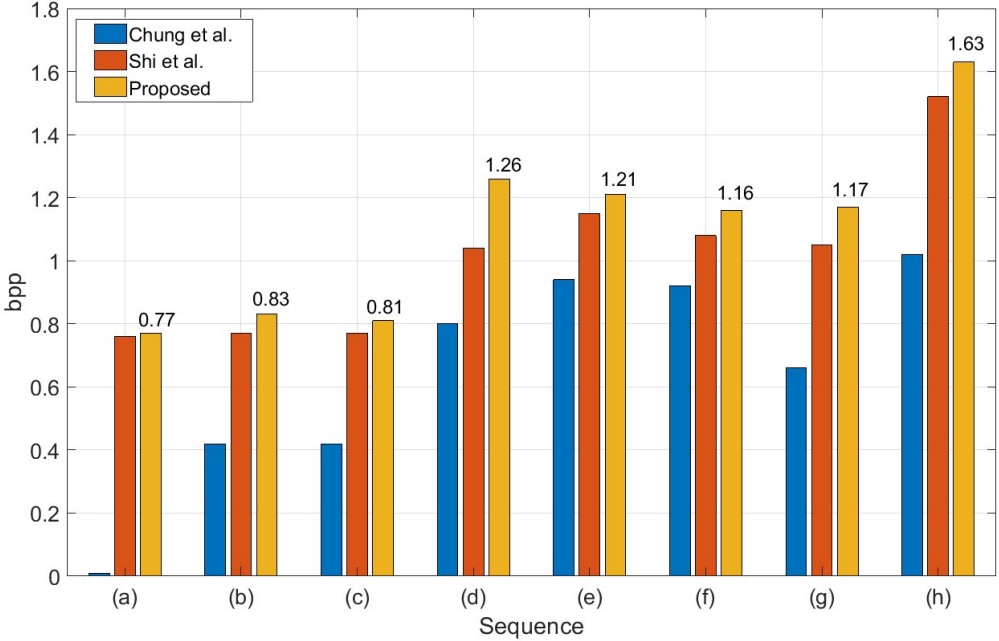

**Figure 6.** Performance comparison of three data hiding methods using eight depth maps.

In Table 1, we compare the performance of the proposed method and the existing methods by using PSNR with various BPPs and the depth map, "Shark". As the embedding rates increases, it appears that Chung et al.'s PSNR is sharply decreased. The reason is that Chung et al.'s method does not use the same method of quality control and the allowable depth ranges in the data embedding procedure. On the other hand, Shi et al.'s and our methods use the allowable pixel ranges and quality control, thus, they have good performance. In addition, our method shows slightly better performance than Shi et al.'s method because it uses an accurate prediction method.

**Table 1.** The comparision of the PSNR at the different BPP for the depth map, "Shark".

| BPP | Chung et al.'s Method PSNR (dB) | Shi et al.'s Method PSNR (dB) | The Proposed Method PSNR (dB) |
|---|---|---|---|
| 0.1 | 59.6257 | 61.1498 | 61.1504 |
| 0.2 | 54.2950 | 58.1325 | 58.1328 |
| 0.3 | 51.6756 | 56.3714 | 56.3744 |
| 0.4 | 49.6793 | 55.1210 | 55.1246 |
| 0.5 | 47.5926 | 54.1483 | 54.1521 |
| 0.6 | 46.6222 | 53.3583 | 53.3613 |
| 0.7 | 45.9716 | 52.6881 | 52.6907 |
| 0.8 | 45.5451 | 52.1081 | 52.1113 |

As shown in Table 2, we know that the maximum BPP of Chung et al.'s method is less than that of both Shi et al.'s and our proposed method. However, since the BPP is low, the PSNR is relatively higher than that of Shi et al.'s and our method.

**Table 2.** PSNR comparison for maximum BPP on each image using various methods.

| Image | Chung et al. | | Shi et al. | | Our Scheme | |
|---|---|---|---|---|---|---|
| | BPP | PSNR (dB) | BPP | PSNR (dB) | BPP | PSNR (dB) |
| (a) Poznan_Hall2 | 0.0050 | 74.5704 | 0.7559 | 49.2263 | 0.7700 | 49.1044 |
| (b) Poznan_Street | 0.4222 | 53.9985 | 0.7713 | 49.3750 | 0.8339 | 48.9424 |
| (c) Undo_Dancer | 0.4215 | 53.6574 | 0.7693 | 49.0931 | 0.8094 | 48.8123 |
| (d) GT_Fly | 0.7956 | 48.6470 | 1.0363 | 44.5942 | 1.2603 | 43.5969 |
| (e) Kendo | 0.9393 | 51.4107 | 1.1485 | 45.6550 | 1.2093 | 45.4096 |
| (f) Ballons | 0.9179 | 51.5102 | 1.0813 | 45.5657 | 1.1633 | 45.2515 |
| (g) Newspaper | 0.6621 | 52.9301 | 1.0533 | 49.0001 | 1.1661 | 48.5315 |
| (h) Shark | 0.9608 | 45.2074 | 1.5194 | 41.0606 | 1.6274 | 40.7411 |

Therefore, if the BPP is the same for the three methods, the PSNRs of Shi et al.'s and our proposed method are better than that of Chang et al.'s method. In "Newspaper", there is the highest difference of PSNR between our proposed and the Shi et al.'s method, because it seems that the depth map is an image including a high-frequency property. Thus, it can be seen that the proposed method had high prediction competence on depth maps with high-frequency characteristics. Moreover, the average BPP of our proposed method is higher (by 0.09) than that of Shi et al.'s method, while our method is less than 0.39 dB compared to that of Shi et al.'s method in the aspect of PSNR. However, in this case, it is indistinguishable from the viewpoint of the usual human visual system. Therefore, it can be recognized that the proposed scheme improves somewhat regarding BPP.

In Figure 7, the control variable $\sigma$ (Equations (12) and (13)) is applied to adjust the embedding capacity and quality of the depth map. Its principle is that the amount of embedding capacity increases in proportion to the value of the variable $\sigma$, while the quality of depth map decreases as $\sigma$ increases. When the control variable $\sigma = 1$, the maximum embedding rate is 0.8 and the PSNR is 52 dB. When the variable $\sigma = 7$, BPP is measured from the lowest 0.1 to the maximum 1.6 and we obtain that the PSNR is from 48.5 dB to 41 dB. Under a strict communication environment, it may be useful to use the control variable for secret communication.

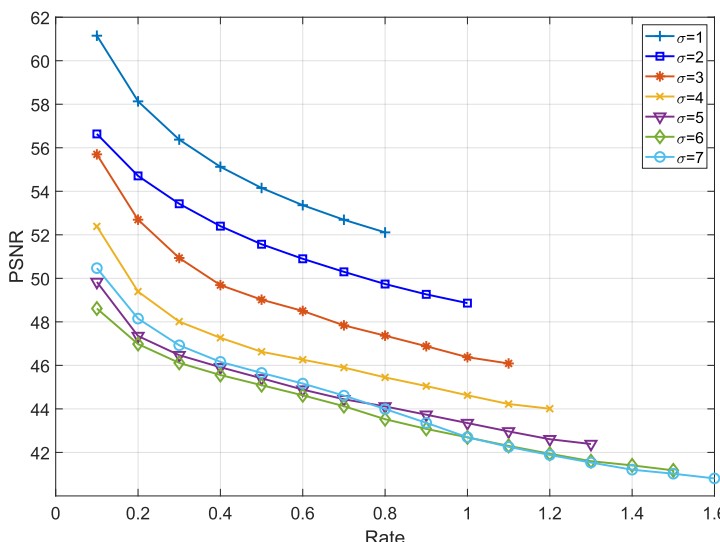

**Figure 7.** The relationship between BPP and PSNR according to control variable $\sigma$ with depth map "Shark".

Table 3 shows PSNRs for the eight depth maps under various BPPs using three methods (two existing and the proposed methods) when $\sigma = 1$. In the table, we can see that depth maps (a), (e), and (f) may hide data up to 0.9 BPP.

**Table 3.** PSNR measurement for the eight depth maps when $\sigma = 1$.

| BPP | PSNRs | | | | | | | |
|-----|-------|-------|-------|-------|-------|-------|-------|-------|
| | **(a)** | **(b)** | **(c)** | **(d)** | **(e)** | **(f)** | **(g)** | **(h)** |
| 0.1 | 61.1483 | 61.1333 | 61.1505 | 61.1533 | 61.1649 | 61.1649 | 61.1421 | 61.1504 |
| 0.2 | 58.1324 | 58.1279 | 58.1374 | 58.1310 | 58.1378 | 58.1378 | 58.1260 | 58.1328 |
| 0.3 | 56.3710 | 56.3680 | 56.3769 | 56.3762 | 56.3735 | 56.3735 | 56.3642 | 56.3744 |
| 0.4 | 55.1215 | 55.1214 | 55.1246 | 55.1220 | 55.1250 | 55.1250 | 55.1171 | 55.1246 |
| 0.5 | 54.1511 | 54.1503 | 54.1547 | 54.1511 | 54.1514 | 54.1514 | 54.1458 | 54.1521 |
| 0.6 | 53.3607 | 53.3599 | 53.3635 | 53.3602 | 53.3635 | 53.3635 | 53.3590 | 53.3613 |
| 0.7 | 52.6905 | 52.6890 | 52.6929 | 52.6900 | 52.6921 | 52.6921 | 52.6886 | 52.6907 |
| 0.8 | 52.1107 | – | 52.1138 | 52.1091 | 52.1116 | 52.1116 | 52.1085 | 52.1113 |
| 0.9 | 51.5992 | – | – | – | 51.6007 | 51.6007 | – | – |

The depth map of (a), (e), and (f) has a higher embedding capacity than the other depth maps; because the sum of pixels $\varphi_n$ (Equation (3)) of these depth maps are high. In other words, there are a number of pixels having a wide range of allowable pixels. From this point of view, it can be seen that sum of pixels $\varphi_n$ of the depth map (b) is low. It is proved that the proposed method maintains very high PSNR like the conventional DHs through various simulations.

Figure 8 is a visual comparison of the marked depth maps generated from the simulation results derived from three methods in "Poznan_Street". In Figure 8, the BPP for (c), (d), and (f) are 0.4222, 0.7713, and 0.8339, respectively, and the PSNRs of these are 53.9985 dB, 49.3750 dB, and 48.9424 dB, respectively. As mentioned above, the embedding rates of the proposed method are about twice as high as that of Chung et al.'s method, and also our proposed scheme provides a good depth map quality about 49 dB.

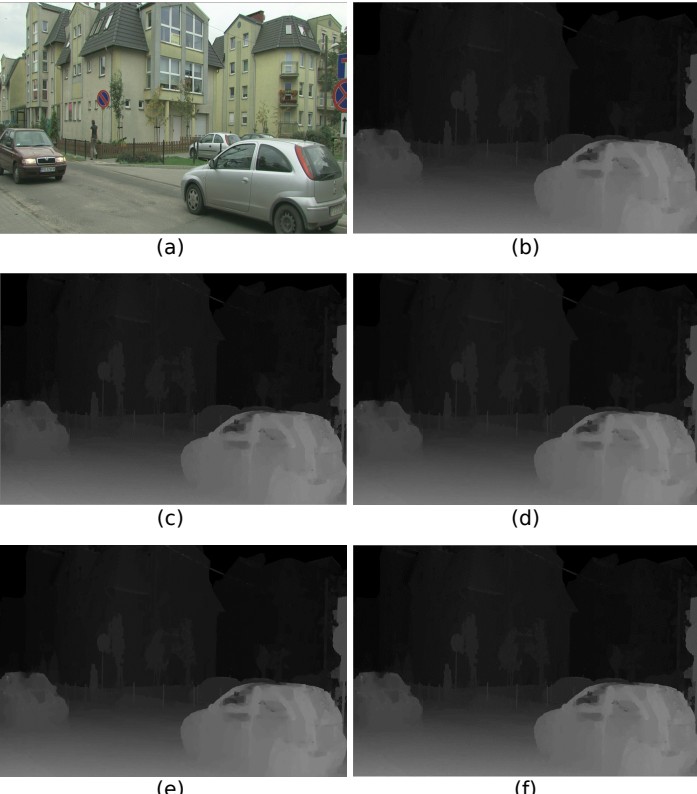

**Figure 8.** The relationship between BPP and PSNR according to control variable, $\sigma$ in depth map on "Poznan_Street".

Figure 9 shows a visual representation of how much pixels are distorted during the data-embedding procedure for the depth map, "Shark". The two marked depth maps by Shi et al.'s and the proposed method include 0.4 BPP. We can easily observe the distortion of the original pixels through the comparison of the histogram made by Shi et al.'s and the proposed method. As shown in the histogram, it seems that the marked histograms are very similar to the original histogram. For this reason, the two marked depth maps show a very high image quality about 55 dB. In graylevel 106 and 107, we may recognize that there is a small difference between the the two methods. As a result, it is proved that our proposed method has fewer errors compared to Shi et al.'s method.

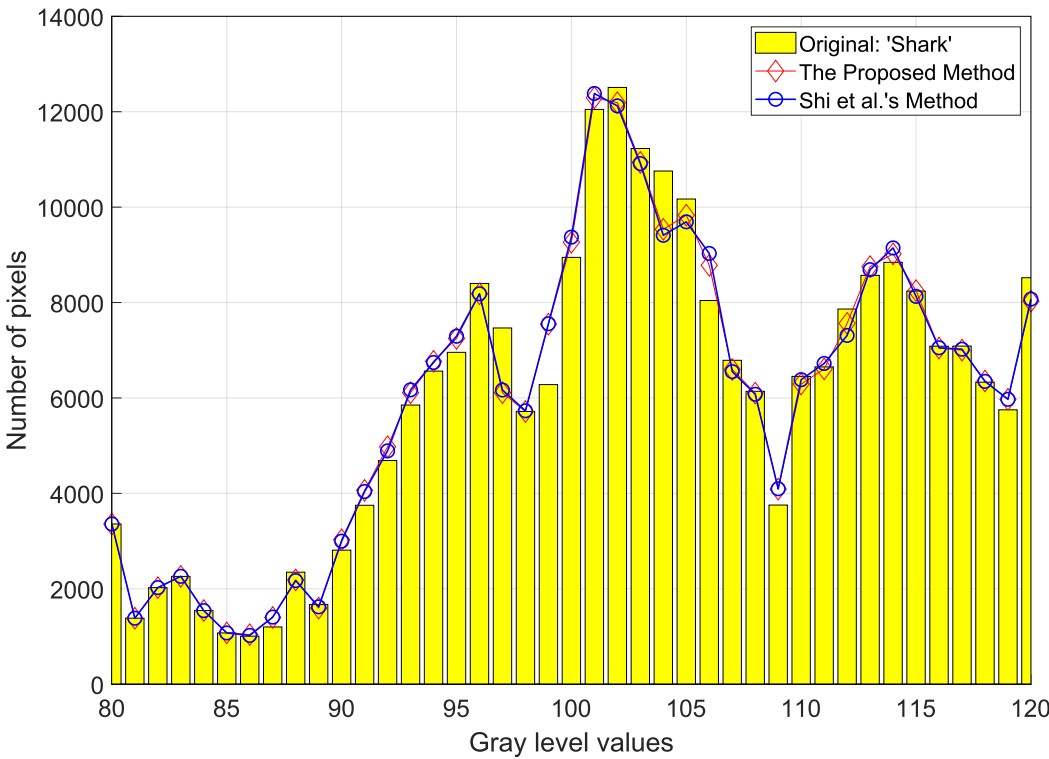

**Figure 9.** Comparison of histogram coefficients on depth map—"Shark" between Shi et al.'s method and the proposed method.

## 5. Conclusions

In this paper, we introduced a method to hide metadata in 3D videos using RDH technology, which is one of many watermarking technologies. The proposed RDH is the PEE method, which hides a large amount of data while minimizing the damage of cover image using LSB of depth map. The accuracy of the pixel prediction is very important to the performance of the PEE method. To improve the accuracy, we proposed an efficient MVD-based RDH using inter-component prediction that predicts depth pixels using MVD-related texture pixels. The newly introduced inter-component prediction may improve the performance of the RDH because the prediction precision is higher than the conventional diamond shape prediction. Especially, the prediction of the depth map in the texture image with high frequency characteristics showed excellent performance. Experimental results demonstrated that the proposed method achieves higher embedding capacity than the all the previous methods by improving the prediction accuracy.

**Author Contributions:** J.Y.L., C.K. conceived and designed the model for research and pre-processed and analyzed the data and the obtained inference. J.Y.L. simulated the design using Visual C ++. C.K. and C.-N.Y. wrote the paper. J.Y.L., C.K., C.-N.Y. checked and edited the manuscript. The final manuscript has been read and approved by all authors.

**Funding:** This research was supported in part by Ministry of Science and Technology (MOST), under Grant 107-2221-E-259-007. This research was supported by the Basic Science Research Program through the National Research Foundation of Korea (NRF) funded by (2015R1D1A1A01059253), and was supported under the framework of international cooperation program managed by NRF (2016K2A9A2A05005255). This work was supported by the National Research Foundation of Korea(NRF) grant funded by the Korea government(MSIT) (No. 2018R1C1B5086072).

**Acknowledgments:** The authors are grateful to the editors and the anonymous reviewers for providing us with insightful comments and suggestions throughout the revision process.

**Conflicts of Interest:** The authors declare no conflict of interest.

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
