# Peer review of "Reversible Data Hiding Using Inter-Component Prediction in Multiview Video Plus Depth"

_electronics, doi:10.3390/electronics8050514_

Round 1

Reviewer 1 Report

The topic is interesting.  Hiding data in an image/video will deeply attracts users.

The introduction and related works are nice and clean, just some graphs in evaluation are too large.

Please proof read it, there are a few typos.  for example " The prediction prediction and the performance of the` ...and I really do not understand prediction prediction of...

In your evaluation, try to do a comparison between larger testing datasets, such as 100 images.  8 images are too small, 

Author Response

Question: (1) The introduction and related works are nice and clean, just some graphs in evaluation are too large.

Response: Depending on your advice, we reduced the size of the images.

Question: (2) Please proof read it, there are a few typos.  for example, " The prediction prediction and the performance of the` ...and I really do not understand prediction prediction of...

Response: Thank you for giving us a good advice. An error occurred entering the word “prediction”. we fixed this error.

Question: (3) In your evaluation, try to do a comparison between larger testing datasets, such as 100 images.  8 images are too small,

Response: Thank you for your valuable feedback. Most data hiding methods provides simulation results using Tables or Charts using standard images. This means that any of the readers (e.g., students, researchers, and professors) can evaluate the reliability of the paper by simulating the results using standard images and the algorithms provided. In addition, since our proposed algorithm provides the simulation of restoring the hidden data using the encoding and decoding method, the evaluation in our paper is very reasonable. For this reason, many existing data hiding methods used about 10 images to perform an experiment on their methods, and to evaluate the performance of those methods. This means that the number of images is not a problem for the performance evaluation of data hiding. Therefore, although we evaluated the performance of data hiding with about 10 images, we are confident that there is no problem in measuring and comparing the performance.

Reviewer 2 Report

Authors proposed a reversible data hiding technique for multiview video and depth map. I recommend to address the following minor comments:

1. Please focus why RDH is important in the Abstract.

2. It would be better if authors briefly introduce other watermarking techniques, such as depth-image based rendering (DIBR) watermarking technique which can be found in [a], before describing reversible data hiding methods for multiview videos in the introduction.

[a] M. Asikuzzaman and M. R. Pickering, "An Overview of Digital Video Watermarking," in IEEE Transactions on Circuits and Systems for Video Technology, vol. 28, no. 9, pp. 2131-2153, Sept. 2018.

3. Authors need to explain more literatures in the Related Work section.

Author Response

Question: (1) Please focus why RDH is important in the Abstract.

Response: Thank you for giving us a good advice. We briefly added an importance of RDH in the Abstract section as follows.

Abstract: With the advent of 3D video compression and Internet   technology, 3D videos have been distributed in the world as well as copyright   infringement cases are often reported. Data hiding is a part of watermarking   technology and has copyright protection capability. In this paper, we use 3D   video as a cover medium for secret communication using reversible data hiding   (RDH) technology. To use of RDH may be an advantage,   because it may recover the cover image after extracting the data from the   cover image, so that it does not affect 3D synthesis. Recently, Chung et al. [26] introduced RDH for   depth map using prediction-error expansion (PEE) and rhombus prediction for   protecting copyright of 3D videos. The performance of Chung et al.'s method   is efficient, but they did not find the way for developing pixel resources to   maximize data capacity. In this paper, we will improve the performance of   embedding capacity using PEE, inter-component prediction, and allowable pixel   ranges. Inter-component prediction utilizes a strong correlation between the   texture image and the depth map in MVD. Moreover, our proposed scheme   provides an ability to control the quality of depth map by a simple formula.   Experimental results demonstrate that the proposed method is more efficient   than the existing RDH methods in terms of capacity.

Question: (2) It would be better if authors briefly introduce other watermarking techniques, such as depth-image based rendering (DIBR) watermarking technique which can be found in [a], before describing reversible data hiding methods for multiview videos in the introduction.

Response: Thank you for giving us a good advice. We briefly added a summary of some papers on watermarking technology related to DIBR as follows.

Asikuzzaman   et al. [21] proposed a digital watermarking   approach that inserts a watermark into the DT CWT coefficients of a central   view. The center left and right views of the DIBR 3D video with that   watermark are protected by this digital watermarking. This method is robust   to most common video distortions, such as lossy JPEG compression and   additional noise, as well as geometric attacks such as scaling, rotation, and   cropping. Pei and Wang [22]   proposed a 3D watermarking technique based on the D-NOSE model which can   detect the suitable region of the depth image for watermark embedding. As   synthesis of the view is very sensitive to variations in depth values, this   scheme focuses mainly on the synthesis error. Wang et al. [23] exploited scale-invariant feature transform (SIFT)-based   feature points to synchronize a watermark but focused on only signal   processing and omitted geometric attacks.

Question: (3) Authors need to explain more literatures in the Related Work section.

Response: We briefly added Subsection (2.4 ~ 2.7) into Section 2 (Related Works) in accordance with your advice.

Reviewer 3 Report

The paper presents a research on reversible data hiding (RDH) improvement using inter-component prediction in multiview video plus depth. The authors propose a data-hiding (DH) method based on prediction-error expansion (PEE). Moreover, reversible DH method is proposed based on a depth no synthesis-error (D-NOSE), PEE and inter-component prediction. The paper is enhanced with numerous examples for methods performance evaluation. 

The authors could improve the abstract and conclusions sections. Specifically, abstract starts with a reference (Chung et al. introduces..)

Author Response

Question: (1) The authors could improve the abstract and conclusions sections. Specifically, abstract starts with a reference (Chung et al. introduces..)

Response: Thank you for giving us a good advice. According to your comment, we rewrote the Abstract and Conclusions.

Abstract: With the advent of 3D video   compression and Internet technology, 3D videos have been distributed in the   world as well as copyright infringement cases are often reported. Data hiding   is a part of watermarking technology and has copyright protection capability.   In this paper, we use 3D video as a cover medium for secret communication   using data hiding technology. To use of reversible data hiding (RDH) may be   an advantage, because it may recover the cover image after extracting the   data from the cover image, so that it does not affect 3D synthesis. Recently,   Chung et al. [26] introduced RDH   for depth map using prediction-error expansion (PEE) and rhombus prediction   for protecting copyright of 3D videos. The performance of Chung et al.'s method is efficient,   but they did not find the way for developing pixel resources to maximize data   capacity. In this paper, we will improve the performance of embedding   capacity using PEE, inter-component prediction, and allowable pixel ranges.   Inter-component prediction utilizes a strong correlation between the texture   image and the depth map in MVD. Moreover, our proposed scheme provides an   ability to control the quality of depth map by a simple formula. Experimental   results demonstrate that the proposed method is more efficient than the   existing RDH methods in terms of capacity.

Conclusions: In this paper, we introduced a   method to hide metadata in 3D video using RDH technology, which is one of   watermarking technologies. The proposed RDH is the PEE method, which hide   large amount of data while minimizing the damage of cover image using LSB of   depth map. The accuracy of the pixel prediction is very important to the   performance of the PEE method. To improve accuracy, we proposed an efficient   MVD-based RDH using inter- component prediction that predicts depth pixels   using MVD-related texture pixels. The newly introduced inter-component   prediction may improve the performance of the RDH because the prediction   precision is higher than the conventional diamond shape prediction.   Especially, the prediction of the depth map in the texture image with high   frequency characteristics showed excellent performance. Experimental results   demonstrated that the proposed method achieves the higher embedding capacity   than the all the previous methods by improving the prediction accuracy.
